# Online Cure Monitoring and Modelling of Cyanate Ester-Based Composites for High Temperature Applications

**DOI:** 10.3390/polym13183021

**Published:** 2021-09-07

**Authors:** Lyaysan Amirova, Christian Brauner, Markus Grob, Nicolas Gort, Fabian Schadt, Nikos Pantelelis, Thomas Ricard, Wilco Gerrits

**Affiliations:** 1Institute for Polymer Engineering, University of Applied Sciences and Arts Northwestern Switzerland, Klosterzelgstrasse 2, 5210 Windisch, Switzerland; markus.grob@fhnw.ch (M.G.); nicolas.gort@fhnw.ch (N.G.); 2Plastics Training and Technology Center (KATZ), Schachenallee 29, 5000 Aarau, Switzerland; fabian.schadt@katz.ch; 3Synthesites SNC, Avenue du Lycée Français 5, 1180 Uccle, Belgium; be@synthesites.com; 4North Thin Ply Technology, Chemin du Closel 3, 1020 Renens, Switzerland; t.ricard@thinplytechnology.com; 5Netherlands Aerospace Centre, Voorsterweg 31, 8316 PR Marknesse, The Netherlands; Wilco.Gerrits@nlr.nl

**Keywords:** cyanate ester, prepreg technology, online cure monitoring, cure kinetics

## Abstract

A cure kinetics investigation of a high temperature-resistant phenol novolac cyanate ester toughened with polyether sulfone (CE-PES blend) was undertaken using non-isothermal differential scanning calorimetry. Thin ply carbon fiber prepreg, based on the CE-PES formulation, was fabricated, and plates for further in-situ cure monitoring were manufactured using automated fiber placement. Online monitoring of the curing behavior utilizing Optimold sensors and Online Resin State software from Synthesites was carried out. The estimation of the glass transition temperature and degree of cure allowed us to compare real time data with the calculated parameters of the CE-PES formulation. Alongside a good agreement between the observed online data and predicted model, the excellent performance of the developed sensors at temperatures above 260 °C was also demonstrated.

## 1. Introduction

Fiber-reinforced polymers (FRPs) are commonly used in the aircraft, automotive, marine, and construction industries [1]. Several molding techniques are used for the manufacturing of FRPs, including Resin Transfer Molding (RTM), Vacuum Assisted Resin Transfer Molding (VARTM), and autoclave molding techniques using prepregs, pultrusion, etc. To obtain a good quality for composite parts, all of these processes require a proper mold design, successful preforming, and a defined cure cycle of the materials. Resin manufacturers usually provide recommendations for the cure cycle, but in the industry this often cannot be accurately performed due to differing additives and mold geometries [2]. Therefore, it is essential to study the curing of the resin first. The characterization, design, and optimization of the cure process will help to control the exotherm and predict the cure cycle of the materials.

Cure cycle optimization includes achieving the target degree of cure, controlling the composite temperature to avoid thermal degradation, and minimizing the residual stresses from a non-uniform temperature distribution during the cure process [3].

There are several methods that allow us to calculate and optimize cure kinetics in idealized laboratory conditions (Differential Scanning Calorimetry DSC, Dynamical Mechanical Analysis DMA, rheology), but only a few methods are suitable for online cure monitoring: infrared, ultrasonic wave propagation, and Raman and dielectric spectroscopy [4,5,6,7,8].

Although extensive research has been devoted to determining the correlation between dielectric properties and the degree of curing [9,10], only laboratory and limited industrial scale applications exist.

*Synthesites* has developed a new process monitoring system based on the direct measurement of resistivity (DC-based), with a focus on industrial manufacturing [11]. The Optimold system directly measures resin resistivity and temperature using specialized sensors capable of the in-situ measurement of the resin’s resistance, from very low values at high temperatures to fully cured resins, with the measured resistance ranging from 105 Ohm up to 1014 Ohm [12].

Cyanate esters (CE) are very attractive thermosetting resins due to their inherently high performance properties, including their excellent dielectric and chemical resistance, as well as high temperature stability [13]. However, due to their high crosslink density, they are very brittle [14].

In a previous work by the EU Project Sustainable and Cost Efficient High Performance Composite Structures demanding Temperature and Fire Resistance (SuCoHS), the toughening of phenol novolac CE using thermoplastic polyether sulfone (PES) has been successfully performed [15]. The improvement in the fracture toughness was 132% for the blend with 20 parts per hundred resins of PES. During the project, thin ply prepreg based on catalyzed toughened CE was developed by *North Thin Ply Technology* (NTPT). The developed system allowed autoclave curing at 180 °C; nevertheless, it requires a post-curing temperature of 260 °C. Standard sensors cannot be employed above 230 °C. In this study the application of a cure monitoring system at high temperatures was demonstrated and applied to this newly developed material system. The curing investigation of CE-based composites was shown by classical methods (DSC analysis, development of a cure kinetics approach) and compared to the developed online process monitoring system.

## 2. Materials and Methods

### 2.1. Materials

The resin used in this study was based on phenol novolac catalyzed cyanate ester (CE, monomer molar mass 381.39 g/mol). This toughened formulation was developed and investigated in a previous work [15] and contains 15 phr (per hundred resin) of polyethersulfone (PES, Sumikaexcel 2603 MP, Sumitomo Chemicals, Tokyo, Japan). Thin-ply carbon fiber prepreg based on the developed toughened cyanate ester and carbon fiber T800S (Toray Corp., Tokyo, Japan) was manufactured by *North Thin Ply Technology* (NTPT, Lausanne, Switzerland). The obtained prepreg had a thickness of 0.06 mm, a fiber volume ratio of 58%, and a fiber areal weight of 67 gsm.

In-situ cure monitoring was performed during the curing of plates made from prepreg tapes with a width of 6.35 mm, using an automated fiber placement (AFP) process on a Coriolis AFP deposition system. The manufacturing and curing of the plates were done in a standard autoclave (Scholz, Coesfeld, Germany) at *Netherlands Aerospace Centre* (NLR), and the curing process was monitored by three Optimold systems from *Synthesites*.

The prepreg manufacturer (NTPT) recommended the following cure cycle [16]. The samples were heated up to 120 °C with a heating rate of 2 °C/min and kept at that temperature for 2 h under a pressure of 5 bar to avoid porosity. A second heating step was performed at 180 °C, with the same ramp of 2 °C/min, maintaining that temperature for 2 h (curing stage). Then, the samples were cooled down with a ramp of 1 °C/min to room temperature. Post-curing was applied in an oven at 260 °C with a heating rate of 1 °C/min, maintained for 2 h to achieve final curing.

### 2.2. Methods

#### 2.2.1. Cure Kinetics Model

The cure kinetics of the used resin were studied using non-isothermal differential scanning calorimetry (DSC). The cure kinetics of cyanate esters have been investigated in many research papers [17,18]. A common approach is to model the curing reaction using a differential equation, according to Karkanas [19] (Equation (1). This consists of the sum of a catalytic (first term) and an autocatalytic (second term) part:(1)$$\frac{dX}{dt}=k_{1}{\left(1-X\right)}^{l}+k_{2}X^{m}{\left(1-X\right)}^{n}$$
where *X* is the degree of curing, parameters *l*, *m,* and *n* represent the reaction order, while *k_i_* is defined as:(2)$$k_{i}=A_{i}\exp\left(\frac{-E_{Ai}}{RT}\right)\;i=1,2$$
where *E_i_* describes the activation energy *E_A_*, and *R* is the universal gas constant (8.314 J·mol^−1^·K^−1^).

In order to take into account of diffusion-controlled mechanisms towards the end of the reaction [20], we propose inserting a maximum degree of cure, *Xmax,* instead of “1”, into the initial model:(3)$$\frac{dX}{dt}=k_{1}{\left(X_{max}-X\right)}^{l}+k_{2}X^{m}{\left(X_{max}-X\right)}^{n}$$

The experimentally derived degree of curing is then calculated, assuming a linear dependence between the measured enthalpy and the degree of curing, as:(4)$$\frac{dX}{dt}=\frac{dH}{dt}\cdot \frac{1}{H_{tot}}$$

The maximum degree of curing was evaluated in a separate isothermal DSC measurement, in which the specimen was cured at a constant temperature. In the second step, the residual curing enthalpy of this specimen was measured by heating it up with a constant heating rate. The determined maximum degree of curing for different isothermal curing conditions was finally parameterized as a function of the temperature, as:(5)$$X_{max}=\frac{1}{1+\exp\left(-T\cdot a_{max}+b_{max}\right)}$$

Then, the glass transition temperature *T_g_* value could be related to the degree of curing using Di Benedetto’s equation [21]:(6)$$T_{g}=T_{go}+\left(T_{g\infty }-T_{go}\right)\cdot \frac{X\lambda }{1-\left(1-\lambda \right)X}$$
where *T_g_*_0_ and *T_g_*_∞_ are, respectively, the glass transition temperatures for uncrosslinked (*X* = 0) and fully crosslinked polymers (*X* = 1), and λ, the fitting parameter, equals 0.16.

Data points for fitting the equation by least square regression were experimentally determined using dynamic DSC measurements from −40 °C to temperatures in the range of 180–320 °C, following the measurement of the degree of curing for each run.

#### 2.2.2. Cure Monitoring

The Optimold system from *Synthesites* measures the electrical resistance and temperature, which can be used for the estimation of viscosity, the glass transition temperature, and the degree of curing using the Synthesites Online Resin State (ORS) software (Brüssel, Belgian). The resin’s electrical resistance was measured through its excitation with a constant voltage (DC), which was applied at the two electrodes of a sensor in contact with the resin. During the early stages of curing, when the resin was still liquid, the electrical resistance could be directly correlated to the viscosity [22,23,24]. Approaching gelation, the viscosity increases rapidly, so that after this point the measured electrical resistance could be directly correlated to the development of the glass transition temperature [25]. In order to use the cure sensors in direct contact with carbon fibers, a new durable sensor was developed. The new sensor, shown in Figure 1, is well suited to integration with a range of high-volume industrial processes, with an outer diameter of 16 mm and an integrated Pt100 RTD temperature sensor.

The Optimold system allows the resin arrival to be determined and the curing level to be monitored qualitatively without calibration. However, for an accurate online estimation of the glass transition temperature, the calibration of the ORS software is required for the specific resin through the correlation of the sensor’s response to the *T_g_* progression of the resin during curing. After this calibration, the ORS software can provide, in real-time, the evolving *T_g_* for any temperature profile, including strongly non-isothermal cases.

Additionally, within this study, the cure disposable sensors were further improved so that they could be used at higher temperatures (above 230 °C), and therefore the curing of cyanate ester resins could be monitored.

## 3. Results

### 3.1. Cure Kinetics Characterization

The total heat of the reaction was determined from DSC experiments by heating from room temperature to 380 °C at various constant heating rates. An example of DSC measurement data for different heating rates is shown in Figure 2. It can be seen that the maximum measured enthalpy shifts to higher temperatures with increasing heating rates. The measured total reaction enthalpies do not depend on the heating rate, as shown in the measurement results of the total enthalpy in Table 1.

As expected, faster heating rates caused a shift of the maximum heat flow to higher temperatures. As a consequence, the onset and offset temperature of the baseline for enthalpy integration shifted. As summarized in Table 1, the measured total enthalpy was found to be around 573 J/g.

The experimental data points of the degree of curing were obtained as explained in equation 4 and subsequently fitted with the presented model. The result of the fitting using a linear regression is shown in Figure 3. It can be seen that the model shows a good flexibility for characterizing the examined curing rates and temperature range.

The fitting enabled us to extract a parameter set of the cure kinetics model, as listed in Table 2.

In order to check the ability of the model to represent isothermal curing with the integrated extension for diffusion-controlled mechanisms, the predicted degree of curing was compared with the measured degree of cures from isothermal DSC experiments. In Figure 4, one can see that a good agreement was achieved for curing temperatures above 180 °C.

The development of the glass transition temperature (*T_g_*) based on the CE-PES system is shown in Figure 5.

The fitting of the Di Benedetto equation 6 resulted in the parameters shown in Table 3.

The performed calculations were applied for the optimization of the recommended cure cycle by the prepreg manufacturer (NTPT). Since the part produced using the presented material was post-cured freestanding in the oven, it was concentrated to ensure that the actual temperature exceeded the current glass transition temperature as little as possible. This was fulfilled by introducing an inflection on the heating ramp at around 580 min.

The optimized cure cycle can be seen in Figure 6.

### 3.2. Online Cure Monitoring

In order to verify the performance of the Optimold system with the specific resin, a series of isothermal trials were performed using a small metal mold and a durable sensor. In Figure 7, the recorded temperature and resistance at 180, 190, 200, 210, and 220 °C are shown. In contrast to the most popular thermoset resins, such as epoxies and unsaturated polyesters, in the CE resin the level of resistance increased with the temperature. However, the measuring range of Optimold was sufficient to measure the resistance until the end of the curing at high temperatures (Figure 7). Based on previous work performed with bismaleimide and epoxy resins [12], it was possible to correlate the measured resistance to the evolution of the glass transition temperature of the toughened CE resin, as provided by the developed kinetic model. Based on this correlation, the Online Resin State (ORS) calibration was developed, enabling the online estimation of the *T_g_*. As can be seen in Figure 8, there was a good agreement between the *T_g_* based on the kinetic model (*T_g_* kinetic) and the *T_g_* estimated by the ORS module (*T_g_* ORS) during curing for a range of isothermal cases.

Following the development of the ORS module for the CE toughened resin, the cure monitoring system was used in the R&D Autoclave and an oven at NLR in Marknesse (NL), where it was possible to use up to three cure sensors—one durable and two disposable sensors for each trial. For the autoclave installation, the sensor cables were passed through the autoclave wall using a specific feedthrough gland provided by Synthesites. Several trials were executed and monitored within the autoclave, as well as within the oven, with CE NTPT prepregs. When comparing the two autoclave trials (called panels 41 and 51), as shown in Figure 9, the high level of repeatability in the curing stage for all materials and systems can be seen. The recorded temperature and resistance between the trials of panel 41 and panel 51 were very similar, including the online estimation of the *T_g_* evolution, as calculated by the ORS module.

As expected, a high level of repeatability was not achieved when the same material was cured in the oven. In Figure 10, two such trials are depicted: case Oven1, in which the same cure cycle was implemented, and case Oven3, where slightly lower temperature dwells and durations were applied, to check the materials’ performance as well as the monitoring system response. As can be seen in Figure 10, the *T_g_* evolution for case Oven3 was differentiated considerably from the Oven1 case, ending approximately 10 °C lower than in the case of Oven1. The results showed that the durable sensor was a very reliable solution, as it functioned correctly during the curing in the oven as well as in the autoclave.

In contrast, the difference with respect to the curing performance between the recommended cure cycles and between the autoclave (Trial 41) and the oven (case Oven1) was not significant, as can be seen in Figure 11. The cure data for the CE toughened panels, cured in the autoclave and the oven, were in good alignment, while the *T_g_* evolution within the autoclave was slightly quicker and higher, mainly due to the faster reach of the 180 °C dwell.

Regarding the comparison between the kinetic model and the ORS *T_g_* estimation, as can be seen in Figure 12, for trial 41, there was a significant difference until a 30% initial degree of cure was used in the kinetic model calculations to introduce the b-stage cure level of the NTPT prepreg.

To check the high-temperature performance of the materials, including the cure monitoring system, the composite plate of trial 51 was post-cured at 262 °C at NLR. Figure 13 shows the data from the disposable cure sensor that was attached to the panel and the estimation of the *T_g_* development from the ORS module.

Thus, the possibility of applying the cure monitoring system to high-temperature cyanate ester formulations was demonstrated.

## 4. Conclusions

The cure kinetics of a new toughened high temperature-resistant cyanate ester system were investigated. New sensors were developed and tested up to temperatures of 260 °C. The Optimold system was successfully used to monitor the curing and to estimate the online evolution of the glass transition temperature and the degree of curing during the processing of the CF/cyanate ester composites. A comparison with the calculated cure kinetics model data showed good agreement. Furthermore, the new high-temperature disposable sensor was successfully integrated during post-curing. The new CF durable sensor, which does not require glass fabric protection to isolate the sensor from the carbon fibers or copper mesh, was preliminarily but successfully tested at Synthesites, but remains to be tested at NLR production facilities.

## Figures and Tables

**Figure 1 polymers-13-03021-f001:**
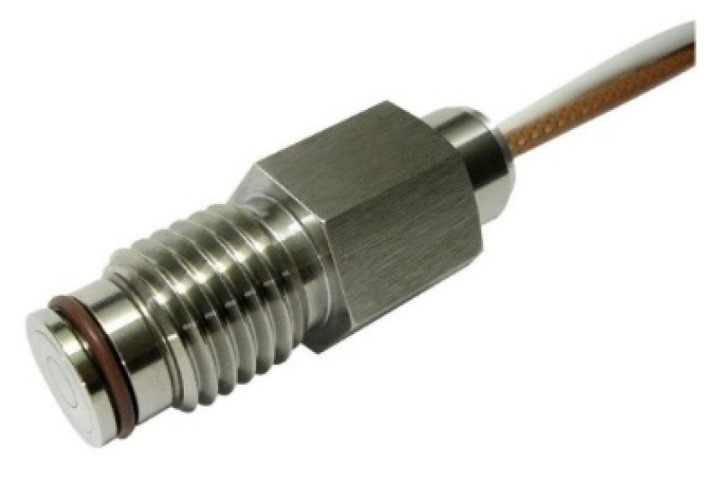
Synthesites CF In-mold cure sensor (size d = 12 mm, length 28 mm).

**Figure 2 polymers-13-03021-f002:**
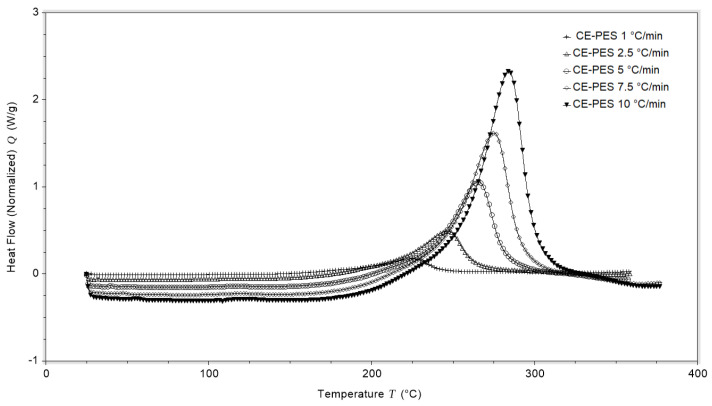
DSC curves for the CE-PES system with different heating ramps.

**Figure 3 polymers-13-03021-f003:**
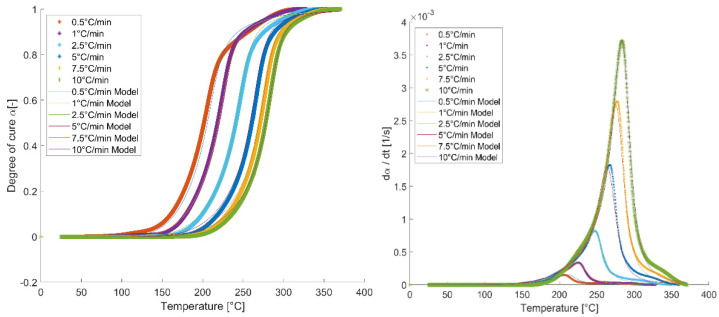
Fitting of the presented cure kinetics model to experimentally obtained data points (**left**: degree of cure vs. temperature, **right**: reaction rate vs. temperature).

**Figure 4 polymers-13-03021-f004:**
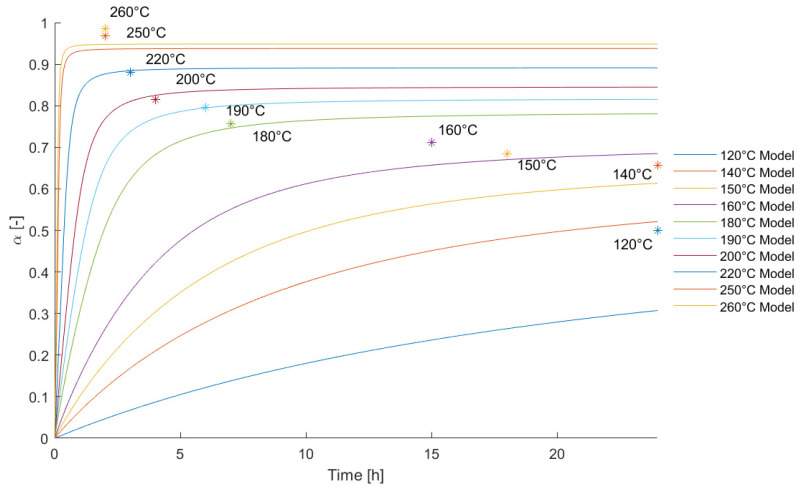
Comparison of the model prediction and measured degree of cures during isothermal curing.

**Figure 5 polymers-13-03021-f005:**
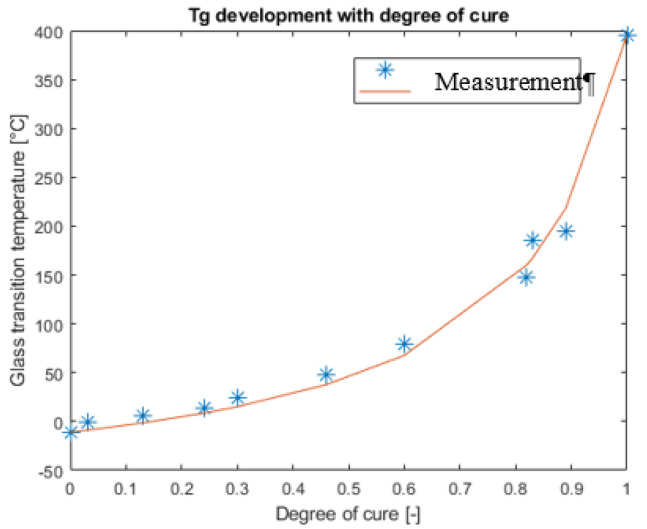
Di Benedetto equation fitted to *T_g_* measurements.

**Figure 6 polymers-13-03021-f006:**
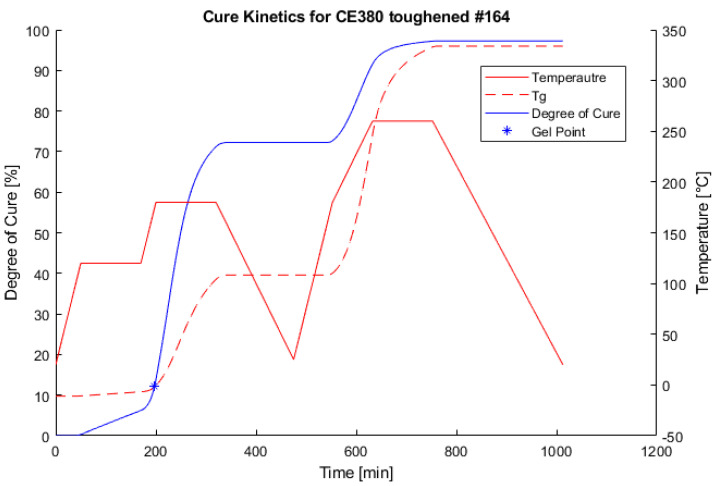
Development of the degree of curing and *T_g_* according to the applied cure cycle.

**Figure 7 polymers-13-03021-f007:**
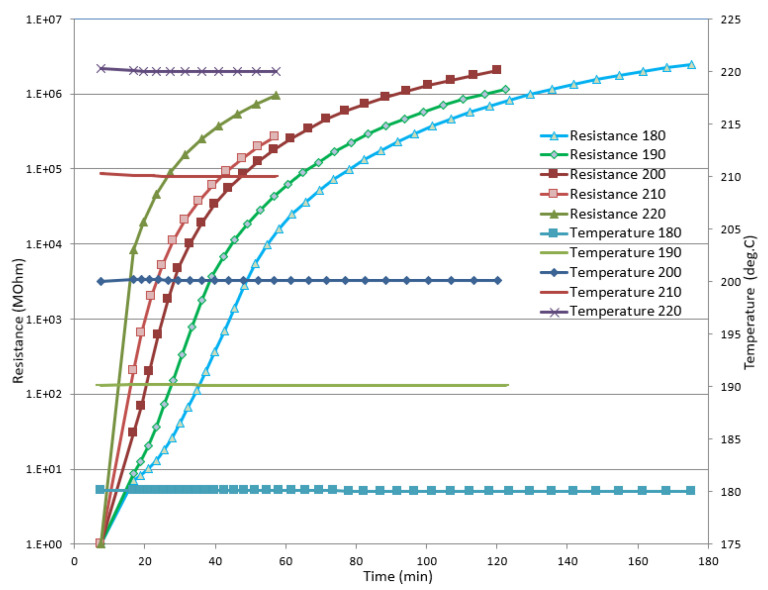
Recorded resistance (left vertical axis) and temperature (right vertical axis) at 180, 190, 200, 210, and 220 °C versus time.

**Figure 8 polymers-13-03021-f008:**
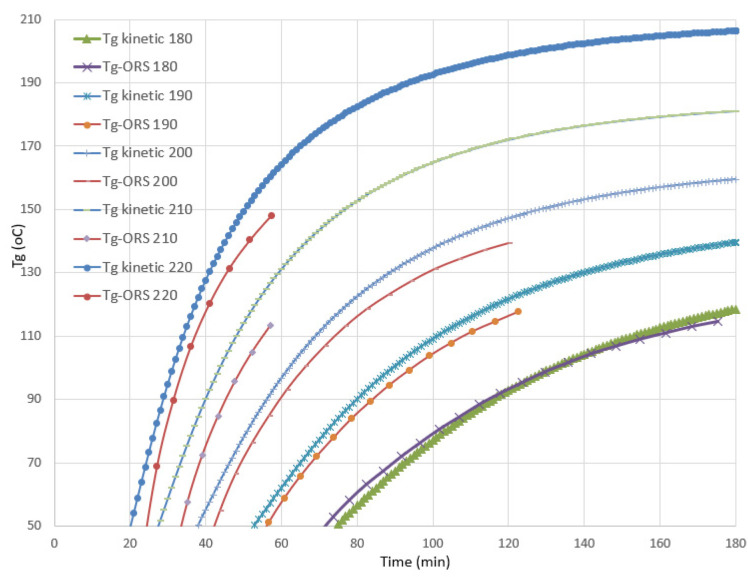
Comparison between the glass transition temperature evolution estimated by the kinetic model (*T_g_* kinetic) and the calibrated cure sensor (*T_g_* ORS) at 180, 190, 200, 210, and 220 °C.

**Figure 9 polymers-13-03021-f009:**
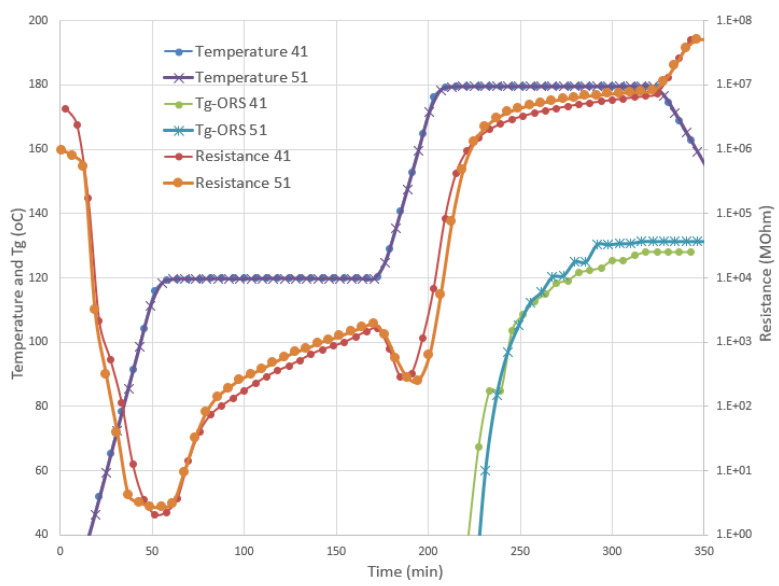
Comparison between the two trials, panel 41 and panel 51, in the autoclave using the durable sensor. The resistance and temperature measurements, as well as the corresponding online *T_g_* estimation, were in good agreement.

**Figure 10 polymers-13-03021-f010:**
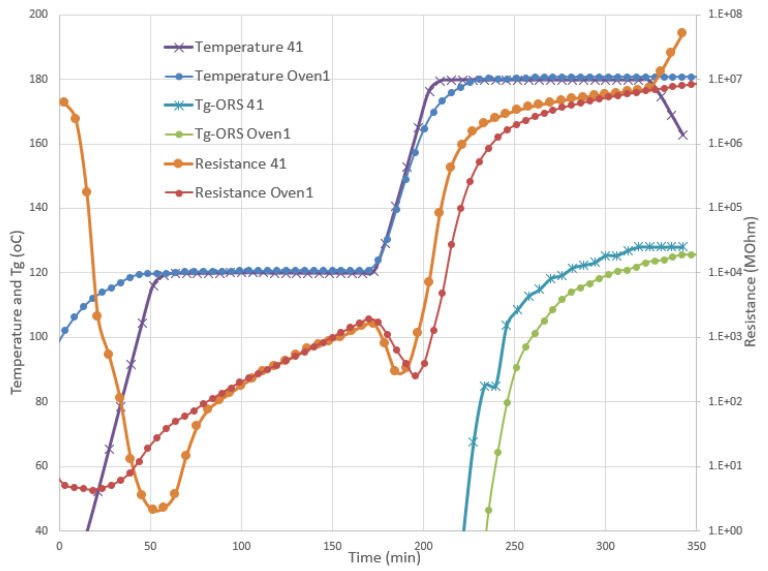
Cases Oven1 and Oven3 cured in the oven: the as-recorded temperature and resistance from the durable sensor were synchronized by the start of the second heating ramp. Shorter and lower curing temperatures resulted in a lower estimated *T_g_* (*T_g_*-ORS Oven3).

**Figure 11 polymers-13-03021-f011:**
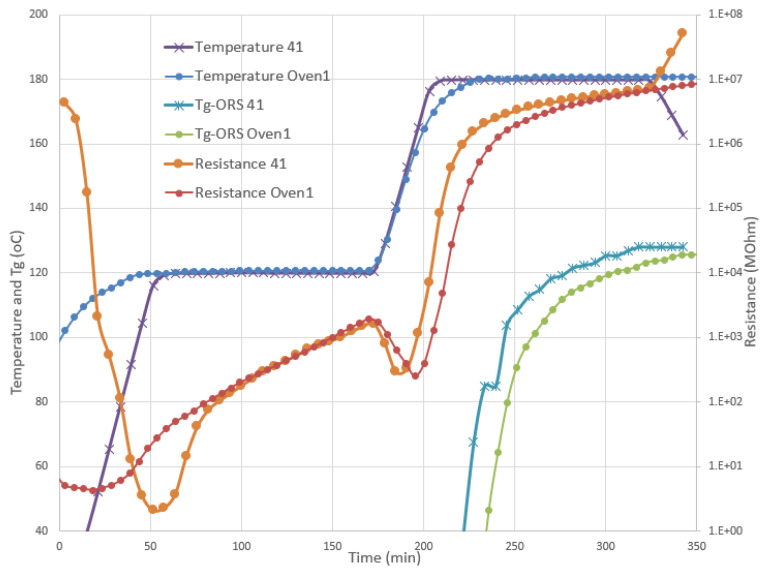
Comparison between the resistance, temperature, and *Tg* estimated with ORS for two curing trials, using the durable sensor in the oven (Oven1) and the autoclave (Trial 41).

**Figure 12 polymers-13-03021-f012:**
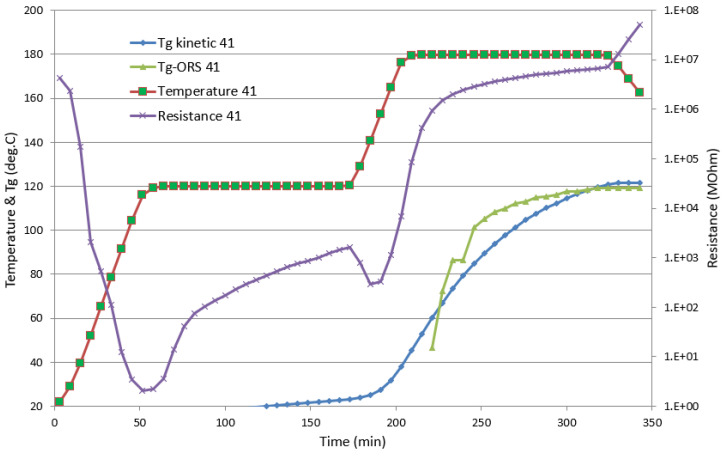
Autoclave Trial 41: Temperature and resistance from the durable sensor as recorded, and the estimated evolution of *T_g_* using the kinetic model (*T_g_* kinetic 41) and the ORS module (*T_g_*-ORS 41).

**Figure 13 polymers-13-03021-f013:**
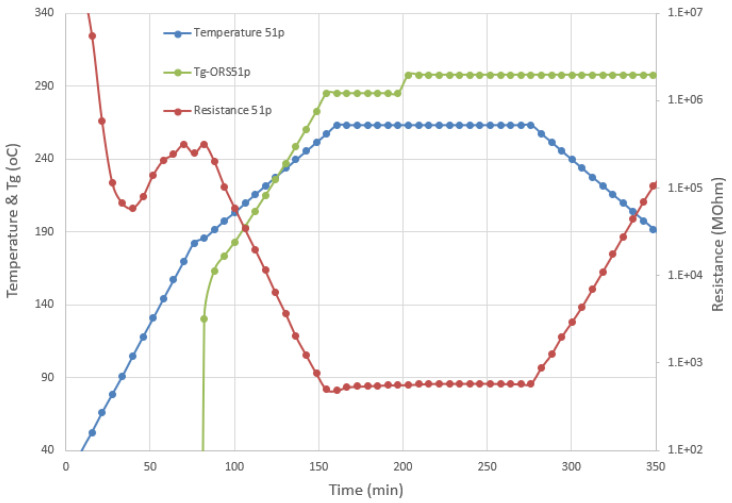
Post-curing of panel 51: Recorded temperature, resistance, and *T_g_* estimation.

**Table 1 polymers-13-03021-t001:** Measured total enthalpies from DSC measurements with a constant heating rate.

Heating Rate [K/min]	Onset [°C]	Offset [°C]	Tot. Enthalpy [J/g]
0.5	120	325	552.8
0.5	118	328	540.1
1	126	336	515.6
2.5	129	350	581.0
5	150	363	571.1
5	144	362	591.2
7.5	157	368	583.1
10	164	371	591.5
		Average	573.0
		Std. Deviation	18.2

**Table 2 polymers-13-03021-t002:** Resulting model parameters from the fitting to experimental data.

Parameter	Value	Unit
A1	1.08 × 10^2^	1/s
E1	5.05 × 10^4^	J/mol
A2	3.40 × 10^6^	1/s
E2	8.38 × 10^4^	J/mol
l	1.68	-
m	1.93	-
n	1.64	-
amax	0.02	1/K
bmax	7.85	-

**Table 3 polymers-13-03021-t003:** Glass transition parameters.

Parameter Name	Parameter Value	Units
λ	0.16	-
*T_g_* _0_	−11	°C
*T_g_* _∞_	395	°C

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
