# Peer review of "Online Cure Monitoring and Modelling of Cyanate Ester-Based Composites for High Temperature Applications"

_polymers, 2021, doi:10.3390/polym13183021_

Round 1

Reviewer 1 Report

  1. Lines 19 to 21 in the abstract are very confusing. I am not sure what the authors are trying to convey. The language has to be corrected.
  2. The introduction is well written but the need for this current study in the backdrop of the existing literature has not been highlighted. The authors should try to present the significance of the present study properly.
  3. It will be good if the authors can represent the curing cycle with a simple schematic for the easy understanding of the reader as this step is the most important step of this study.
  4. The sensor shown in figure 1 is developed by the authors or the company? It is not clear from the write-up.
  5. The authors have mentioned that some improvements have been done to the disposable sensors to be used at elevated temperatures. So, the sensor in Figure 1 has been redesigned or improved? Please clarify.
  6. Figure 2: It will be good if the heating rates are also given in terms of 0C/min rather than in K/min for clarity purposes.
  7. On what basis were these heating rates selected?
  8. In figure 3, what is CpM? Is it 0C/min? If it is then why this sudden shift from K/min to CpM? Please clarify.
  9. Line 164, something is wrong. Please correct.
  10. I would like to know as to why did the authors not select 2 0C/min (which is being used in the actual curing cycle) of heating rate to see the behavior and correlation of the experimental and theoretical models?
  11. A schematic of the experimental setup as described in lines 208 to 218 would be helpful.
  12. Lines 256 to 257: The authors abruptly claim that the sensors can be used to high temperature measurement. However, they fail to give the description of the improvements that were made to the design etc. This should be clearly given with proper explanation and figures as this is one of their major achievements.
  13. It will be good if the authors can compare the performance of the newly developed sensors at higher temperatures with other sensors available in the literature.

Author Response

Dear reviewer,

We would like to thank you for your valuable inputs which have helped us to improve the article. Please find below the point by point response.

  1. Lines 19 to 21 in the abstract are very confusing. I am not sure what the authors are trying to convey. The language has to be corrected.

Corrected and the English was improved using the MDPI language service

  1. The introduction is well written but the need for this current study in the backdrop of the existing literature has not been highlighted. The authors should try to present the significance of the present study properly.

In the introduction we highlight the importance of cure monitoring for high temperature resistant systems. Specialty of this work is application of developed sensors for cure monitoring of carbon fibre composites at temperatures up to 260°C. (Stae of the art only limited to max 200°C)

  1. It will be good if the authors can represent the curing cycle with a simple schematic for the easy understanding of the reader as this step is the most important step of this study.

The cure cycle is presented in Fig 6. The figure was modified to:see article

  1. The sensor shown in figure 1 is developed by the authors or the company? It is not clear from the write-up.

Cure monitorins Optimold system and sensors are developed by the company Synthesites, represented by Nikos Pantelelis who is also co-author of the paper.

  1. The authors have mentioned that some improvements have been done to the disposable sensors to be used at elevated temperatures. So, the sensor in Figure 1 has been redesigned or improved? Please clarify.

Actually both sensors were improved, the disposable one for higher temperatures and the durable one for direct use with carbon fibres. In Figue 1 the durable sensor is presented.

  1. Figure 2: It will be good if the heating rates are also given in terms of 0C/min rather than in K/min for clarity purposes.

Corrected

  1. On what basis were these heating rates selected?

Normally, for non-isothermal cure kinetic study several heating rates in the region between 1 and 15 °C/min are used. In the case of cyanate ester based resin, characterized with high exothermic reaction at around 280°C (for 10°C/min) and degradation above 400°C, it made no sense to exceed 10°C/min, since the end of curing peak merged with the degradation start. Therefore, we focused on lower heating rates since it was also applicable for industrial curing cycle. A typical heat rate in a autoclave is 1 to 2 °C/min

  1. In figure 3, what is CpM? Is it 0C/min? If it is then why this sudden shift from K/min to CpM? Please clarify.

CpM is also heating ramp, it is corrected to °C/min

  1. Line 164, something is wrong. Please correct.

The capture is explaining both graphs in Fig 3. The format and explanation was adapted

  1. I would like to know as to why did the authors not select 2 0C/min (which is being used in the actual curing cycle) of heating rate to see the behavior and correlation of the experimental and theoretical models?

For the cure kinetic study, multiple heating rates were selected to be valid for several curing conditions. This study allows to transform data into isothermal modes to predict the degree of cure under various conditions. The cure cycle included several isothermal steps recommended by NTPT was applied for online cure monitoring and performed cure kinetic investigation helped to understand the development of the glass transition temperature and degree of cure during this cycle (Fig 6).

  1. A schematic of the experimental setup as described in lines 208 to 218 would be helpful.

Unfortunately, there were no photos of the final setup in the autoclave. Unless can be noted, that thermocouples and several sensors were installed in different part of the plate manufactured out of the described prepreg material.

  1. Lines 256 to 257: The authors abruptly claim that the sensors can be used to high temperature measurement. However, they fail to give the description of the improvements that were made to the design etc. This should be clearly given with proper explanation and figures as this is one of their major achievements.

The development of the sensors is done by Synthesites, details of modification stays as company’s Know How.

  1. It will be good if the authors can compare the performance of the newly developed sensors at higher temperatures with other sensors available in the literature.

Due to the fact that available sensors will vail above 200°C we do not see a benefit of this investigation beside the outcome that they will not measure properly.

Reviewer 2 Report

The authors studied the curing kinetics of a high temperature resistant cyanate ester system and developed new sensors for testing. The work is interesting, though some revisions are needed prior to publications.

 1) Scale bar of Fig. 1 is missing.

2) In Figure 6, Tg and Temperature are both colored in red, which is confusing. They should be colored in different colors.

3) As a polymer-based manuscript, the authors should at least show the chemical structures, molecular weight, and viscosity for the precursors and polymers.

Author Response

We would like to thank you for your valuable inputs which have helped us to improve the article. Please find below the point by point response.

 The authors studied the curing kinetics of a high temperature resistant cyanate ester system and developed new sensors for testing. The work is interesting, though some revisions are needed prior to publications.

  • Scale bar of Fig. 1 is missing.

Fig 1 is only a digital photo, the sensor is approximately 28 mm long. It was added in the figure description 

  • In Figure 6, Tg and Temperature are both colored in red, which is confusing. They should be colored in different colors.

Corrected

  • As a polymer-based manuscript, the authors should at least show the chemical structures, molecular weight, and viscosity for the precursors and polymers.

It was changed to: The resin used in this study was based on phenol novolac catalyzed cyanate ester (CE, monomer molar mass 381.39 g/mol). This toughened formulation was developed and investigated in a previous work [15], and contains 15 phr (per hundred resin) of polyeth-ersulfone (PES, Sumikaexcel 2603 MP, Sumitomo Chemicals).

Detailed information of the development of this formulation is given in our previous work

  1. Amirova, L.; Schadt, F.; Grob, M.; Brauner, C.; Ricard, T.; Wille, T. Properties and structure of high temperature resistant cyanate ester/polyethersulfone blends using solvent-free toughening approach. Polym. Bull. 2020 (in press). https://doi.org/10.1007/s00289-020-03493-w.

Round 2

Reviewer 1 Report

None

Author Response

thank you for your input

Reviewer 2 Report

Accept 

Author Response

thank you for your input